# Vasopressin in the Amelioration of Social Functioning in Autism Spectrum Disorder

**DOI:** 10.3390/jcm8071061

**Published:** 2019-07-19

**Authors:** Mohamed A. Hendaus, Fatima A. Jomha, Ahmed H. Alhammadi

**Affiliations:** 1Department of Pediatrics, Section of Academic General Pediatrics, Sidra Medicine, Doha 26999, Qatar; 2Department of Pediatrics, Section of Academic General Pediatrics, Hamad Medical Corporation, Doha 3050, Qatar; 3Department of Clinical Pediatrics, Weill-Cornell Medical College, Doha 26999, Qatar; 4School of Pharmacy, Lebanese International University, Khiara 146404, Lebanon

**Keywords:** autism, functioning, social, vasopressin

## Abstract

Autism spectrum disorder (ASD) is a developmental disability described by diagnostic criteria that comprise deficits in social communication and the existence of repetitive, restricted patterns of behavior, interests, or activities that can last throughout life. Many preclinical studies show the importance of arginine vasopressin (AVP) physiology in social functioning in several mammalian species. Currently, there is a trend to investigate more specific pharmacological agents to improve social functioning in patients with ASD. Neurobiological systems that are crucial for social functioning are the most encouraging conceivable signaling pathways for ASD therapeutic discovery. The AVP signaling pathway is one of the most promising. The purpose of this commentary is to detail the evidence on the use of AVP as an agent that can improve social functioning. The pharmacologic aspects of the drug as well as its potential to ameliorate social functioning characteristics in human and animal studies are described in this manuscript. AVP, especially in its inhaled form, seems to be safe and beneficial in improving social functioning including in children with autism. Larger randomized studies are required to implement a long awaited safe and feasible treatment in people with a deficiency in social functioning.

## 1. Introduction

Autism spectrum disorder (ASD) is a developmental disability described by diagnostic criteria comprised of deficits in social communication and the existence of repetitive, restricted patterns of behavior, interests, or activities that can last throughout life [1]. According to the World Health Organization (WHO), the worldwide prevalence of ASD is 0.62% [2], but it is higher (1.68%) in some countries such as the United States [3]. ASD exists in all ethnic, racial, and socioeconomic groups, and it is four times more common among boys than girls [4]. Genetics might play a role in acquiring ASD. The literature indicates that the concurrence rate of ASD can range from 36 to 95% in identical twins and 0–31% in non-identical twins [5,6,7,8], and the recurrence rate can reach up to 10% [9,10]. (https://www.cdc.gov/ncbddd/autism/data.html#references). Furthermore, children who are born prematurely, with low birth weight, and those born to older parents are at a higher risk for having ASD [4]. 

ASD tends to occur more often in people who have certain developmental, psychiatric, neurologic, genetic, or chromosomal conditions [2]. Approximately 10% of children with autism also have tuberous sclerosis, Down syndrome (https://www.cdc.gov/ncbddd/birthdefects/downsyndrome.html), fragile X syndrome (https://www.cdc.gov/ncbddd/fxs/facts.html), or other genetic disorders [11,12,13,14].

The core pathophysiology of ASD is not fully understood and therefore it is currently diagnosed based on behavioral criteria [15]. Despite the widely available tools validated to diagnose ASD, the complexity of these measurements might obscure the clear identification of this phenomenon as a behavioral health diagnosis [16].

According to the Center for Disease Control and Prevention (CDC), 44% of children diagnosed with ASD have average to above average intellectual ability, while others might require permanent care and support [4]. Moreover, and unfortunately, patients with ASD might be subject to stigma due to their behavior [2].

The contemporary pharmacological agents for ASD include antidepressants, antipsychotics, and psychostimulants, aimed at symptoms such as hyperactivity, aggression, self-injurious behavior, impulsivity, stereotypies, mood disorders, and anxiety [17].

Currently, there is a trend to investigate more specific pharmacological agents to improve social functioning in patients with ASD. Neurobiological systems that are crucial for social functioning are conceivably the most encouraging signaling pathways for ASD therapeutic discovery. The arginine vasopressin (AVP) signaling pathway is one of the most promising ones [18]. Many preclinical studies have shown the importance of AVP physiology in social functioning in several mammalian species [19,20,21,22].

In 2002, Born et al. [23] studied the effect of inhaled neuropeptides on neurobehavioral functions. The investigators used three peptides (melanocortin, vasopressin, and insulin) via the intranasal route and found that they attained direct access to the cerebrospinal fluid (CSF) within 30 minutes, bypassing the bloodstream. The study included 37 healthy adults (nine females, 27 males, 25–41 years of age). Five patients received 10 mg of inhaled melanocortin, and four patients received 5 mg. Five patients received 80 IU of inhaled AVP and four patients received 40 IU. As for inhaled insulin, eight patients received 40 IU. The rest of the participants received an intranasal placebo. The level of each peptide was measured in the CSF and serum within 80 minutes of the inhaled administration. The result of the study showed statistically significant peptide accumulation in the CSF within 80 min after the administration of melanocortin (10 mg), AVP (80 and 40 IU), and insulin (40 IU), as compared to pre-administration baseline levels and to levels in subjects who were administered sterile water as a placebo. For melanocortin and insulin, peak levels were achieved within 30 minutes after administration, while for AVP, CSF concentrations continued to increase for up to 80 minutes after administration.

## 2. Experimental Section

A search in PubMed, Embase, and Google Scholar databases was conducted using different combinations of the following terms: arginine, vasopressin, social, functioning, autism, and treatment. Moreover, the references of the identified articles were searched for further articles. We also attended conferences and shared information with investigators. Abstracts and titles were inspected, and studies that were appropriate to the topic of interest were selected. Finally, the search was restricted to manuscripts that were published in Spanish and English from inception until July 2016.

## 3. Results

### 3.1. Arginine Vasopressin (AVP)

AVP is a peptide made of nine amino acids that is synthesized in the magnocellular neurons of the hypothalamic paraventricular and supraoptic nuclei as a prohormone “pre-pro-vasopressin”. It is then metabolized to pro-vasopressin prior to arriving to the posterior pituitary along the neuronal axons and is eventually converted to active vasopressin releasing neurophysin-II and co-peptin. It is delivered into the bloodstream by axon terminals in the posterior pituitary [24]. It takes approximately 1–2 h for the whole process of production, transportation, and storage of vasopressin to take place [25].

In addition to its very well-known functions, AVP also plays a crucial role in organizing social behaviors such as social recognition and sexual behavior [24,26].

Repetitive behaviors are probably influenced by AVP since it is considered to be a neuromodulator, targeting the amygdala, hippocampus, striatum, hypothalamus, and nucleus accumbens [19,27]. Irregularities of AVP in ASD can be attributed to different reasons, such as a decrease in hormonal levels due to reduction in synthesis, changes in hormonal delivery, receptor abnormalities [28], or an alteration of the glia and/or accelerated synaptic shearing [29].

### 3.2. Molecular Component of AVP

The genes which code for the production of AVP are located on chromosome 20p13 [20]. AVP has three G-protein receptors (*AVPR1A*, *AVPR1B*, and *AVPR2*), and social effects are mediated through *AVPR1A* [15,30].

### 3.3. Animal Studies

In animal models, there is no consensus regarding whether AVP or AVP antagonists ameliorate behavior or to some extent social functioning.

The role of AVP in individual social memory processing was initially found in male and female rats and male mice [30,31,32,33]. Several studies were conducted in animal models using different routes of administration of AVP. Acute intranasal AVP (IN-AVP) administration in animal models has shown different results on pro-social effects [34,35]. Ramos et al. [36] studied the effects of inhaled vasopressin on sociability, body temperature, and heart rate in rats. The investigators were able to administer inhaled AVP in rats by nebulizing the peptide (1 mL of a 5 or 10 mg/mL solution) into a small enclosed chamber over a 2 min period. The study showed that the group of rats who received nebulized AVP (5 or 10 mg/mL) showed decreased ano-genital sniffing and increased social proximity (adjacent lying) in the social interaction test. In terms of physiologic changes, rats showed some sort of bradycardia and hypothermia (body temperature: *F* (1,7) = 89.71, *p* < 0.001; heart rate: *F* (1,7) = 17.10, *p* < 0.01). Plasma AVP levels were increased 10 min after nebulized AVP, showing levels above those seen with a behaviorally effective injected dose of AVP (0.005 mg/kg intraperitoneal injection).

Simmons et al. [37] studied the effect of intranasal vasopressin in partner preference in male prairie voles. The investigators recruited 103 prairie vole subjects (52 males, 51 females). Each participant was randomly assigned to one of four groups (three interventional and one control). The first three groups received a low dose of AVP (0.05 IU/kg), a medium dose of AVP (0.5 IU/kg), and a high dose of AVP (5.0 IU/kg), while the control group received normal saline. The investigators found no effect of intranasal AVP on anxiety or social-related behaviors in males. However, it did affect play behavior in females (*F* (3,50) = 2.750; *p* = 0.05).

Other animal studies have used different routes of AVP administration other than inhalation. For instance, one study showed that prenatal AVP injections have an effect on fetal suckling behavior [38]. Early life stress might have an effect on AVP release. Lukas et al. [39] investigated whether early life stress hinders social recognition behavior in Wistar rats. The investigators found that the lack of social recognition in rats with maternal separation was associated with a deficiency in AVP release from the lateral septum as compared to the rats in the control group. The researchers eventually conducted retrodialysis of AVP (1 μg/mL, 3.3 μL/min, 30 min) into the lateral septum which resulted in social memory restoration after 60 min.

### 3.4. Clinical Studies

Thus far, there have been few human studies that deal with vasopressin and autism. Investigators at the University of Stanford are currently conducting a double-blind, randomized, placebo-controlled, parallel design testing the efficacy and tolerability of a 4 week intranasal AVP treatment in a sample of *n* = 30 children with ASD, aged 6–12 years. The study [40] was presented in the World Summit of Pediatrics in Madrid, Spain in 2018 and is registered under clinical trials with identifier NCT01962870 [41]. Children aged 6 to 9.5 years received a dose of 24 IU (12 IU twice daily). Participants aged 9.6 to 12 years received a dose of 32 IU (16 IU twice daily). The preliminary results of the study show that the social responsiveness scale (a parent-report measure) is better in children treated with AVP compared to the placebo group. In addition, social improvement was documented by the children’s performance-based measures and by the health care provider’s impression. The investigators also reported decreased restricted and repetitive behaviors as well as anxiety in the treatment group. In terms of the serum AVP concentration, individuals with the highest pre-treatment blood concentrations benefitted the most from AVP (but not placebo) administration. The authors disclosed the safety profile of the medication. There was no significant difference in adverse events between the placebo and treatment groups. Moreover, there were no significant changes from baseline in vital signs, anthropometric measurements, electrolytes, and electrocardiogram between the two groups.

In Maryland, Zink et al. [42] studied the effect of vasopressin on social recognition. The investigator recruited 20 right-handed, healthy male participants aged 18–43 years. The participants were randomized to self-administer 40 UI of vasopressin or placebo intranasally. A functional magnetic resonance imaging (fMRI) of the brain and an implicit social recognition matching task were conducted 45 minutes after the administration of vasopressin. The investigators concluded that vasopressin induced a regionally specific alteration on the left temporo-parietal junction, establishing a neurobiological mechanism for prosocial neuropeptide effects in humans. Moreover, a year earlier Zink et al. [43] conducted a study with the same methodology, revealing changes in connectivity between the subgenual cingulate and supragenual cingulate with the administration of vasopressin.

Parker et al. [44] presented the safety and efficacy of 4 week intranasal AVP administration to improve social abilities in children with ASD using a double-blind, randomized, placebo-controlled trial design. The investigators recruited 22 children aged 6 to 12 years. Participants were randomized 1:1 to receive either AVP treatment (12 IU BID or 16 IU BID based on age) or placebo treatment for 4 weeks. The study showed that participants who received AVP improved by an average of 12.8 ± 3.5 points on the SRS-2 (Social Responsiveness Scale) Total Score (*p* = 0.0102), but participants who received the placebo did not have significant improvement in the SRS-2 total scores. Moreover, children with a higher *AVPRv1a* gene expression showed greater improvement in SRS-2 total scores. The investigators did not notice any difference in the rates of adverse events between the treatment and placebo group.

In Australia, Guastella et al. [45] assessed the effect of intranasal AVP in encoding happy and angry faces in 48 male adult students from the University of Sydney (age = 18–60 years, mean = 21.98, SD = 7.04). The participants were randomly assigned in a double-blind manner to receive either 20 IU of AVP (*n* = 24) or a placebo (*n* = 24). The investigators presented 54 happy, angry, or neutral human faces to the group of participants. One day later, participants were asked to make “remember”, “know”, or “new” judgments for a blend of 108 new and previously seen faces. The study showed that compared to the group of people who received the placebo, the recipients who inhaled AVP were more likely to make “know” judgments for previously seen happy and angry faces.

AVP can have different effects on opposite sexes. Thompson et al. [46] studied the sex-specific influences of vasopressin on human social communication. The study included 20 women and 18 men who were all matched with control participants. Using double-blind conditions, a 1 mL solution containing 20 units of AVP was administered via intranasal spray over 2 minutes. The results of the study showed that in males, AVP stimulates agonistic facial motor impression when reacting to the faces of unfamiliar men and diminishes perceptions of the amicability of those faces. On the other hand, in females, AVP stimulates affiliative facial motor impression when reacting to the faces of unfamiliar women and augments perceptions of the amicability of those faces.

The role of AVP in human risky cooperative behavior was investigated by Brunnlieb et al. [47]. The investigators studied 59 healthy adult males who received nasal spray with either 40 IU of AVP or placebo 30 min before the instigation of cooperative behavior. In two double-blind experiments, AVP increased willingness to cooperate and make decisions with financial consequences in the “Stag hunt” cooperation game. The study showed that AVP downregulated the blood oxygen level-dependent activity (BOLD signal) in the left dorsolateral prefrontal cortex (dlPFC), a risk-integration region, and augmented the left dlPFC functional connectivity with the ventral pallidum, an AVP receptor-rich social reward processing region.

Wu et al. [48] studied the differential effects of intranasal vasopressin on the processing of adult and infant cues. The study included 48 males aged 18–26 years (mean age: 22.46, SD = 2.02). Participants were randomly assigned to receive vasopressin (*n* = 24) or placebo (*n* = 24) treatment. After intranasal and double-blind treatment of AVP (20 IU) or placebo, participants were requested to rate their subjective willingness to advance towards infant and adult faces in specific contexts briefed by cue words while an electroencephalogram was recording. The study showed that AVP administration increased approaching ratings to neutral and positive other-gender adult faces compared to emotional matched same-gender adult faces. Furthermore, compared to placebo treatment, intranasal AVP led to quick attentiveness to both adult and infant cues, whereas AVP treatment only assisted sustained attention to infant cues but not adult cues. The authors hypothesized that these results could be due to differential roles of AVP in the processing of adult- and infant-related cues.

Price et al. [49] conducted a study to determine if AVP promotes different responses to same- and other-sex faces in men, and if those effects are dose-dependent. The investigators included 86 male subjects between the ages of 18 and 30 years. The study compared the effects of two doses of AVP previously used in intranasal studies, 20 and 40 IU, on subjective ratings of same- and other-sex faces in human males 50 min after drug or placebo administration and again after a few or several days (between 2 and 20 days after drug delivery). The study concluded that intranasal AVP effects on face processing may vary depending on the dose.

Finally, Rilling et al. [50] delineated the effects of AVP on subjective judgments and neural responses to same- and other-sex faces in males and females. The study randomized 40 healthy men and 40 healthy women to receive either 40 IU intranasal AVP or a saline placebo approximately 30 min before imaging their brain function with fMRI as they viewed same- and other-sex faces. Participants were also scanned a second time a few days later with no treatment to appraise the persistence of AVP effects over time. The study showed that AVP increased positive ratings of same-sex faces in women, with indication that these effects endured until the second scan. Despite AVP showing no effects on same-sex ratings in men, AVP increased positive ratings of same-sex faces several days later. AVP inflection of brain function was concentrated on the nucleus accumbens (NAc) and the lateral septum, which are considered as two reward processing areas involved in the formation of social bonds. Participants who received AVP showed increases in the right NAc and bilateral lateral septum responses to female faces among men. Finally, the authors hypothesized that intranasal AVP influences subjective ratings and neural responses to same- and other-sex faces in men and women.

### 3.5. Discussion

In vertebrates, the AVP system innervates preserved and broadly distributed neural networks that adjust social information managing and behavior [51]. AVP signaling can contribute to social impairments in children with ASD, and modern genetic studies have assisted in proving this. Moreover, the human genes for AVP-neurophysin II (NPII) are mapped on chromosome 20p13, and it has been noticed that multiple genetic loci within the AVP region may influence the development of ASD [52], including childhood aggression [53].

Intranasal vasopressin has been used to test the neuropeptide on different groups of individuals [54]. Table 1 summarizes the studies that have used intranasal vasopressin to ameliorate social impairment. Low intranasal vasopressin doses can have pleasant effects such as improvement in the social responsiveness scale, enhancement of the prosocial effect, improvement in the judgment of faces, stimulation of facial motor impression, downregulation of the risk-integration region, accelerated attention to both adult and infant cues, and increased positive ratings of same-sex faces. However, there were some limitations to the above studies including the sample and power of study. Nevertheless, a study with reasonable power but using a different route of AVP administration showed improvement in the personal and social skills of individuals. In 2017, at the International Meeting for Autism Research (IMFAR), Bolognani et al. [55] presented the results of a randomized, double-blind, placebo-controlled study, Vasopressin Antagonist to Improve Social Communication in Autism (VANILLA), which investigated the efficacy and safety of a V1a antagonist (RG7314) in adult men with ASD. The study included 223 adult participants and was a multi-center, double-blind, placebo-controlled dose-escalation study using three doses (1.5, 4, and 10 mg administered orally). The treatment duration was 12 weeks. The investigators concluded that drug was safe and, despite SRS not showing a significant effect, the Vineland Adaptive Behavior Scales improved for the 4 and 10 mg doses.

Guiding resources toward the AVP system may possibly result in the discovery of critical biomarkers of ASD, permitting earlier diagnosis, intervention, and eventually a better prognosis [56].

### 3.6. Safety Profile

There were no significant side effects reported in clinical studies [41,42,43,44,45,46,47,48,49,50,55]. Common complications of vasopressin when co-administered with a moderate to high dose of norepinephrine were documented. In addition, vasopressin can have cardiac complications such as ventricular arrhythmias, coronary ischemia, myocardial infarction, and severe hypertension when administered by methods other than inhalation [57]. In terms of gastroenterological side effects, there have been reports of gastroentero ischemia leading to bowel necrosis. Furthermore, bronchospasm, urticarial, hyponatremia, angioedema, rashes, peripheral vasoconstrictions, and local irritation at the injection site have been reported with non-inhaled vasopressin [58].

## 4. Conclusions

Arginine vasopressin, especially in its inhaled form, seems to be safe and beneficial in improving social functioning, including in children with autism. Larger randomized studies are required to implement a long-awaited safe and feasible treatment in people with deficiency in social functioning.

## Figures and Tables

**Table 1 jcm-08-01061-t001:** List of studies in which inhaled vasopressin was used.

Study	Dose of Inhaled Vasopressin	Results
Hardan et al. [41]	24 IU ^^^ or 32 IU BID ^#^	improvement in social responsiveness scale
Zink et al. [42]	40 IU once daily	prosocial effect
Parker et al. [44]	12 IU or 16 IU BID	improvement in social responsiveness scale
Guastella et al. [45]	20 IU once daily	improvement in judgment of faces
Thompson et al. [46]	20 IU once daily	stimulates facial motor impression
Brunnlieb et al. [47]	40 IU once daily	downregulated risk-integration region
Wu et al. [48]	20 IU once daily	quick attention to both adult and infant cues
Price et al. [49]	20 IU or 40 IU once daily	improvement in effects on face processing
Rilling et al. [50]	40 IU once daily	increased positive ratings of same-sex faces

^^^ International Unit; ^#^ Twice a day.

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
