# Peer review of "Vasopressin in the Amelioration of Social Functioning in Autism Spectrum Disorder"

_jcm, 2019, doi:10.3390/jcm8071061_

Round 1
Reviewer 1 Report
Dysregulation of vasopressin pathway has been implicated in the manifestation and maintenance of abnormalities seen in ASD. The review by Hendaus et al. tried to summarize preclinical and translational studies related to this important field of study. Unfortunately, the review is simply the summary of the published (but not comprehensive) works, which reminiscent several of published review articles. Most of the paragraphs are inevitably discontinuous and broken into pieces and it’s hard to obtain new information and directions for the future study from the review.
Forgeot d'Arc et al. Tinkering with the vasopressin pathway in autism. Sci Transl Med. 2019 May 8;11(491).
Cataldo et al. A Review of Oxytocin and Arginine-Vasopressin Receptors and Their Modulation of Autism Spectrum Disorder. Front Mol Neurosci. 2018 Feb 13;11:27.
Iovino et al. The Role of Neurohypophyseal Hormones Vasopressin and
Oxytocin in Neuropsychiatric Disorders. Endocr Metab Immune Disord Drug Targets. 2018;18(4):341-347.
Johnson and Young. Oxytocin and vasopressin neural networks: Implications
for social behavioral diversity and translational neuroscience. Neurosci Biobehav Rev. 2017 May;76(Pt A):87-98.
Author Response
Reviewer 1:
Dear Sir or Madam,
Thank you for taking the time and effort to review our manuscript. Your comments have added high value to our paper.
I am attaching two new versions of the manuscript: 1. with track changes and 2. Clean
As you can notice in the version with track changes, I have done major revisions and I did my best to fulfil the requests of the two reviewers without affecting the thoughts/comments of either.
As for your comments:
Comments and Suggestions for Authors
Dysregulation of vasopressin pathway has been implicated in the manifestation and maintenance of abnormalities seen in ASD. The review by Hendaus et al. tried to summarize preclinical and translational studies related to this important field of study. Unfortunately, the review is simply the summary of the published (but not comprehensive) works, which reminiscent several of published review articles. Most of the paragraphs are inevitably discontinuous and broken into pieces and it’s hard to obtain new information and directions for the future study from the review.
As seen in the new manuscripts, some information was added, other was deleted and some of the paragraphs were shuffled to make flow better.
Forgeot d'Arc et al. Tinkering with the vasopressin pathway in autism. Sci Transl Med. 2019 May 8;11(491).
Cataldo et al. A Review of Oxytocin and Arginine-Vasopressin Receptors and Their Modulation of Autism Spectrum Disorder. Front Mol Neurosci. 2018 Feb 13;11:27.
Iovino et al. The Role of Neurohypophyseal Hormones Vasopressin and
Oxytocin in Neuropsychiatric Disorders. Endocr Metab Immune Disord Drug Targets. 2018;18(4):341-347.
Johnson and Young. Oxytocin and vasopressin neural networks: Implications
for social behavioral diversity and translational neuroscience. Neurosci Biobehav Rev. 2017 May;76(Pt A):87-98.
The above suggested citations were added per your recommendations.
Please let us know if you have any questions or concerns.
Thank you
Sincerely,
The authors

Reviewer 2 Report
2 ½ pages reviewing the basics of ASD. Don’t think necessary.
Describe a thoughtful systematic review protocol
Spend another 2 1/2 pages describing basic chemistry and use of vasopressin for other disorders unrelated to ASD. This is not tied into ASD.
Line 146 does not make sense “higher level of allocating more money [41]” on its own.
The authors randomly report studies of vasopressin antagonists and agonists (supplemental vasopressin) with no discussion of why both seem to be of benefit.
There is also no discussion of why vasopressin agonists or antagonists might work differently from oxytocin and for boys or girls.
There is no Discussion or summary and critical part of this review, just a listing of studies with few of them related to ASD.
The Hardan study is reported a coming from the University of Standford (not Stanford).
The manuscript needs editing for grammar and careless errors of leaving words edited out.
All of these “mistakes” raise concerns about the rigor of this manuscript preparation.
Author Response
Reviewer 2:
Dear Sir or Madam,
Thank you for taking the time and effort to review our manuscript. Your comments have added high value to our paper.
I am attaching two new versions of the manuscript: 1. with track changes and 2. Clean
As you can notice in the version with track changes, I have done major revisions and I did my best to fulfil the requests of the two reviewers without affecting the thoughts/comments of either.
As for your comments:
Comments and Suggestions for Authors
2 ½ pages reviewing the basics of ASD. Don’t think necessary.
Describe a thoughtful systematic review protocol
Spend another 2 1/2 pages describing basic chemistry and use of vasopressin for other disorders unrelated to ASD. This is not tied into ASD.
As seen in the new manuscripts, some information was added, other was deleted and some of the paragraphs were shuffled to make flow better.
Line 146 does not make sense “higher level of allocating more money [41]” on its own.
The authors randomly report studies of vasopressin antagonists and agonists (supplemental vasopressin) with no discussion of why both seem to be of benefit.
As seen in the new manuscripts, some information was added, other was deleted and some of the paragraphs were shuffled to make flow better.
There is also no discussion of why vasopressin agonists or antagonists might work differently from oxytocin and for boys or girls.
As seen in the new manuscripts, some information was added, other was deleted and some of the paragraphs were shuffled to make flow better.
There is no Discussion or summary and critical part of this review, just a listing of studies with few of them related to ASD.
A discussion part was added
The Hardan study is reported a coming from the University of Standford (not Stanford).
Done . Thank you
The manuscript needs editing for grammar and careless errors of leaving words edited out.
All of these “mistakes” raise concerns about the rigor of this manuscript preparation.
Corrected . Many Thanks
Please let us know if you have any questions or concerns.
Many Thanks
The authors

Round 2
Reviewer 1 Report
Again, I don't think the revised manucript is qualified as a meaningful review on top of already published articles by others.
Reviewer 2 Report
The authors made a very cursory and rapid revision mainly consisting of deleting large portions of text and some very minor word substitution. The Discussion has been significantly improved. The authors have still not addressed the perplexing finding that both vassopressin agonists and antagonists improve autism.